# Harnessing many-body spin environment for long coherence storage and high-fidelity single-shot qubit readout

George Gillard [1✉], Edmund Clarke [2] & Evgeny A. Chekhovich [1✉]

There is a growing interest in hybrid solid-state quantum systems where nuclear spins, interfaced to the electron spin qubit, are used as quantum memory or qubit register. These approaches require long nuclear spin coherence, which until now seemed impossible owing to the disruptive effect of the electron spin. Here we study InGaAs semiconductor quantum dots, demonstrating millisecond-long collective nuclear spin coherence even under inhomogeneous coupling to the electron central spin. We show that the underlying decoherence mechanism is spectral diffusion induced by a fluctuating electron spin. These results provide new understanding of the many-body coherence in central spin systems, required for development of electron-nuclear spin qubits. As a demonstration, we implement a conditional gate that encodes electron spin state onto collective nuclear spin coherence, and use it for a single-shot readout of the electron spin qubit with >99% fidelity.

[1] Department of Physics and Astronomy, University of Sheffield, Sheffield S3 7RH, UK. [2] Department of Electronic and Electrical Engineering, University of Sheffield, Sheffield S1 3JD, UK. ✉email: g.gillard@sheffield.ac.uk; e.chekhovich@sheffield.ac.uk

The original approach to solid-state spin qubits considered the many-body environments, such as nuclear spins, as a decoherence source[1–3]. On the other hand, nuclear spins offer uniquely long coherence storage, making them attractive as buffer memories in photonic quantum information processing systems[4]. Thus, the attention shifted to hybrid systems, where nuclear spins are interfaced with the electron spin qubit[5,6] and used as quantum information processing resource[7–10]. Several material systems are of interest. In group-IV semiconductors, such as diamond and silicon, isotopic purification offers electron and nuclear spin coherence on the order of seconds[11] and hours[12], respectively. The states of nuclear spins, adjacent to the electron spin qubit, are individually addressable and preserve coherence for tens of milliseconds[5,13], but the readout speed and fidelity are restricted by the tiny nuclear magnetic moments. In group-III–V semiconductor quantum dots (QDs) all nuclei have spins that can be probed using efficient optical techniques[14]. The electron spin can be controlled on a picosecond timescale[15] and coherently interfaced to nuclear spin single-quantum excitations[6]. On the other hand, preserving nuclear spin ensemble coherence in presence of the electron central spin proved challenging[16], and, due to the many-body nature, the underlying physics remain an open problem.

The central spin model is described by the hyperfine Hamiltonian $\hat{\mathcal{H}}_{hf} = \sum_j A_j \hat{\mathbf{s}} \cdot \hat{\mathbf{I}}$, where constants $A_j$ ($1 \le j \le N$) characterize the coupling of the single-electron spin $\mathbf{s}$ to the spins $\mathbf{I}_j$ of $N$ lattice nuclei (Fig. 1a). Magnetic interactions between the nuclei with pairwise coupling constants $b_{j,k}$, together with inhomogeneity in $A_j$, result in complex dynamics[2,3,17–19], characteristic of many-body quantum phenomena[20,21]. In a typical epitaxial device (Fig. 1a) a Fermi reservoir of electrons is introduced, and electric field is applied through the gate voltage $V_G$, to control the charge state of the QD. In an empty (0e) InGaAs QD the direct nuclear-nuclear interactions limit the nuclear spin coherence time to a few millisecond range[16,22] $T_{2,N}^{(0e)} \propto h / \max(|b_{j,k}|)$, where $h$ is Planck's constant. The spin of a single electron (1e) induces hyperfine shifts in precession frequencies, ranging from 0 for distant nuclei to $\max(|A_j|)/(2h) \approx 100$ kHz for nuclei at the center of the QD. These shifts, known as Knight shifts, result in short nuclear spin dephasing time $T_{2,N}^{*,(1e)} \propto 2\hbar/\sqrt{\langle A_j^2 \rangle} \approx 3$ μs, determined simply by inhomogeneity of $A_j$ ($\hbar = h/(2\pi)$ is the reduced Planck's constant). Dephasing is reversible and can be refocused with spin echo[23] to reveal the timescale $T_{2,N}$ of irreversible decoherence. Unlike with $T_{2,N}^*$, predicting $T_{2,N}$ is a difficult problem. Previous experiments[16] gave rather short coherence $T_{2,N}^{(1e)} \approx 20$ μs, that was ascribed to indirect nuclear-nuclear interactions mediated by the electron spin. However, to fully understand the QD coherence, one must take into account the effects of the electron Fermi reservoir.

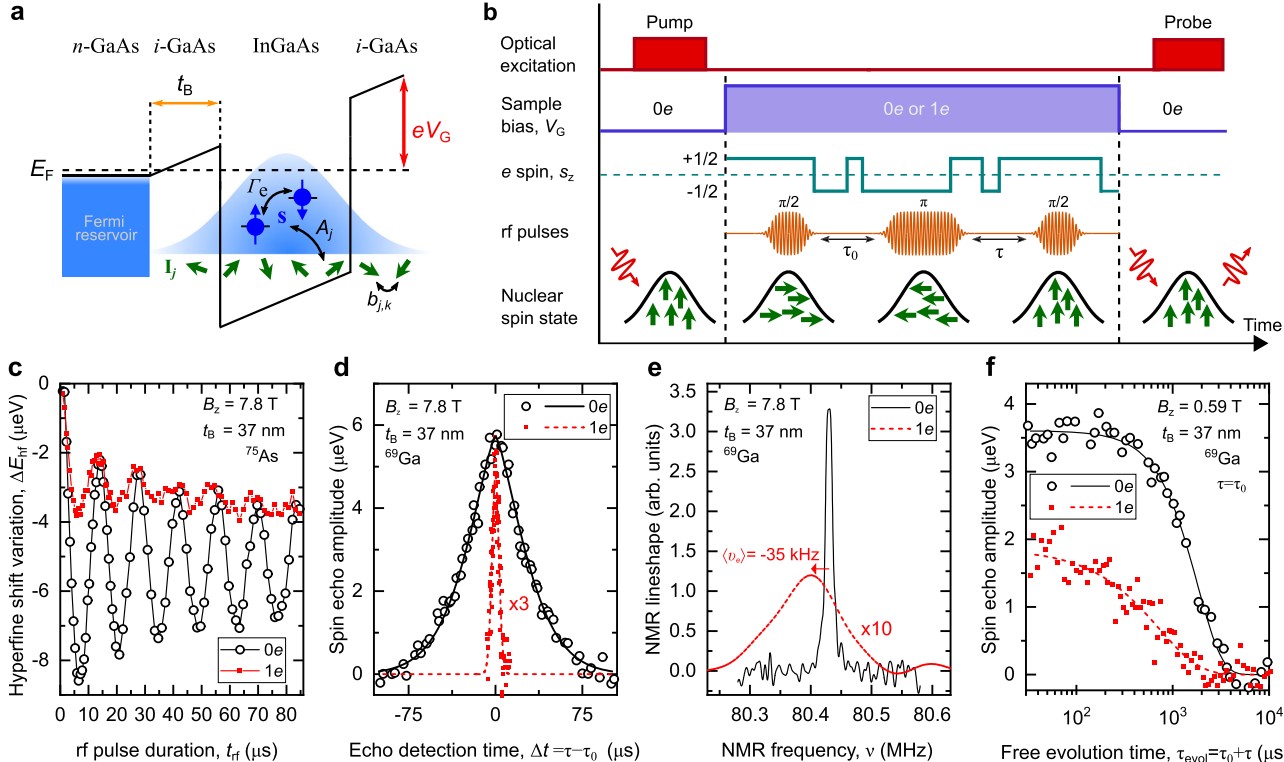

**Fig. 1 Coherent control of nuclear spins coupled to quantum dot electron spin qubit. a** Schematic of a conduction band edge in an n-i-Schottky diode structure containing InGaAs QDs, separated from the electron Fermi reservoir by a tunnel barrier of thickness $t_B$. Sample gate bias $V_G$ is adjusted with respect to Fermi energy $E_F$ to achieve single-electron (1e) QD charging. The electron spin $\mathbf{s}$ (ball and arrow) couples to the nuclear spin ensemble $\mathbf{I}_j$ (small arrows) via inhomogeneous hyperfine interaction. Relaxation between electron spin-up (↑, $s_z = +1/2$) and spin-down (↓, $s_z = -1/2$) states is characterized by rate $\Gamma_e$. **b** NMR experiment timing diagram showing optical pump and probe pulses, used to polarize and measure polarization of the nuclear spins, respectively. Radio-frequency (rf) pulses implementing coherent nuclear spin control at an arbitrary $V_G$ are sketched for spin echo sequence $(\pi/2)_x - \tau_0 - (\pi)_x - \tau - (\pi/2)_x$. The electron spin $s_z$ undergoes random transitions between its two states. **c** Rabi oscillations of the $^{69}$Ga nuclear spins induced by a resonant rf pulse of variable duration $t_{rf}$ in an empty (0e, circles) and charged (1e, squares) QD in the $t_B = 52$ nm sample at $B_z = 7.8$ T. Lines are a guide to the eye. **d** Spin echo evolution as a function of the second delay $\tau$ in the $t_B = 37$ nm sample at $B_z = 7.8$ T, revealing free induction decay in 0e (circles, $\tau_0 = 150$ μs) and 1e states (squares, $\tau_0 = 7.5$ μs, data multiplied by 3). Lines show compressed exponential fits used to derive the nuclear spin dephasing times $T_{2,N}^*$. **e** Fourier transform of (**d**) showing spectral broadening and the average Knight shift $\langle \nu_e \rangle \approx -35$ kHz induced by equilibrium electron spin polarization. Data for 1e is multiplied by 10. **f** Spin echo decay measured by varying the total free evolution times $\tau_{evol} = \tau_0 + \tau$ at $\tau_0 = \tau$. Lines show fitting used to derive the nuclear spin coherence times $T_{2,N}$.

By optimizing the tunnel coupling between QD and the Fermi reservoir, we achieve here an isolated-qubit regime where the lifetime of a QD electron central spin is long and therefore spectral diffusion[24,25] is reduced. We find that indirect nuclear-nuclear interactions are negligible, and inhomogeneous interaction with the central spin is harmless under these conditions, allowing us to achieve long coherence $T_{2,N}^{(1e)} > 1$ ms in a many-body nuclear spin ensemble. While many-body spin environment is often seen as a hindrance for qubits, here we design a protocol where nuclear spin coherence is harnessed for a single-shot readout of the electron spin qubit, with fidelity matching the state of the art[26–32].

## Results

**Coherence of a spin ensemble coupled to the central spin.** We study diode structures (Fig. 1a) with tunnel barrier $t_B = 37$ or 52 nm, thick enough to ensure long single-electron spin lifetimes $T_{1,e}$ up to $\approx 1$ ms or $\approx 1$ s, respectively[33]. We investigate spin-3/2 nuclei of $^{75}$As, $^{69}$Ga and $^{71}$Ga. With magnetic field $B_z$ applied along the sample growth axis, nuclear states with spin projections $I_z = \pm 1/2$ form effective spin-1/2 ensembles that we focus on, whereas $I_z = \pm 3/2$ states are spectrally detuned and can be ignored (see Methods). Hyperfine interaction between the electron spin and nuclear spins[14] (see Supplementary Note 1) provides a tool both for hyperpolarisation of the nuclei via optical pumping, and for detection of the nuclear ensemble polarization via hyperfine shifts $E_{hf} = \sum_j A_j \hat{s}_z \langle \hat{I}_{z,j} \rangle$ in the optical emission spectra of the QD electrons (here $\langle \hat{I}_{z,j} \rangle$ is the expectation value for the $z$ projection spin operator of the $j$-th nuclear spin). The reciprocal hyperfine effect of the electron spin is characterized by the Knight shifts in the nuclear magnetic resonance (NMR) frequencies.

We examine the collective coherence of the QD nuclear spins using optically detected NMR protocol (Fig. 1b). Figure 1c shows the result of an NMR experiment, where a single resonant radio-frequency (rf) burst of duration $t_{rf}$ induces coherent Rabi oscillations between the $I_z = -1/2$ and $+1/2$ nuclear spin states. The inhomogeneous Knight shifts $\nu_{e,j} \propto A_j$ induced by the electron (1e) result in faster damping of coherent nuclear spin rotations compared to an empty QD (0e). To characterize this inhomogeneity we use a spin echo sequence (Fig. 1b), where the initial $\pi/2$ pulse transforms nuclear spin polarization into collective (multi-quantum) coherence, followed by free evolution over time $\tau_0$, a refocusing $\pi$ pulse, and a further free evolution time $\tau$, before the final $\pi/2$ pulse converts the remaining coherence back into optically detectable nuclear spin polarization. The width of the spin echo peak observed around $\tau = \tau_0$ (Fig. 1d) is proportional to dephasing time, which reduces from $T_{2,N}^{*,(0e)} \approx 35$ μs in a neutral QD to $T_{2,N}^{*,(1e)} \approx 4.3$ μs when QD is charged with a single electron. Using Fourier transform, the dephasing dynamics reveal the inhomogeneous broadening of the nuclear spin resonance (Fig. 1e). In an empty QD the broadening of $\approx 13$ kHz is due to the second order quadrupolar shifts, whereas in presence of an electron the $\approx 126$ kHz linewidth is dominated by the inhomogeneous Knight shifts.

By fixing $\tau = \tau_0$ we remove the dephasing and examine pure nuclear spin decoherence by measuring spin echo amplitude decay with increasing total free evolution time $\tau_{evol} = \tau_0 + \tau$ (Fig. 1f). From exponential fitting we find the coherence time $T_{2,N}^{} \approx 1.8$ ms in a bare $^{69}$Ga nuclear spin ensemble, in good agreement with previous studies[16,22]. By contrast, the coherence time $T_{2,N}^{(1e)} \approx 0.7$ ms in presence of a single QD electron is a factor of $\approx 30$ longer than reported earlier[16]. By varying the gate bias $V_G$ (Fig. 2a), we observe a clear plateau in $T_{2,N}$ around $V_G \approx 0.42$ V,

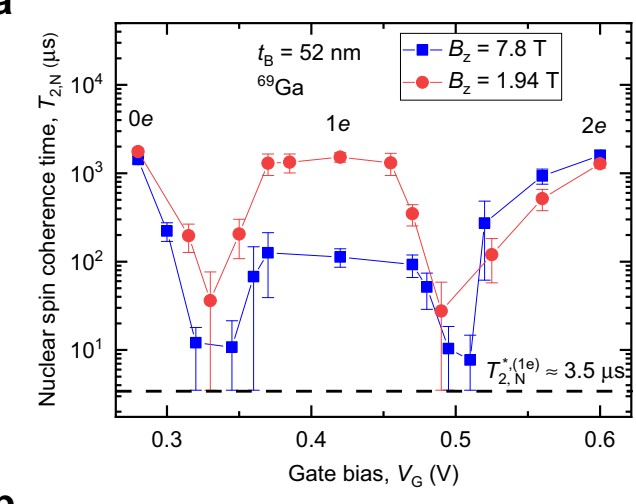

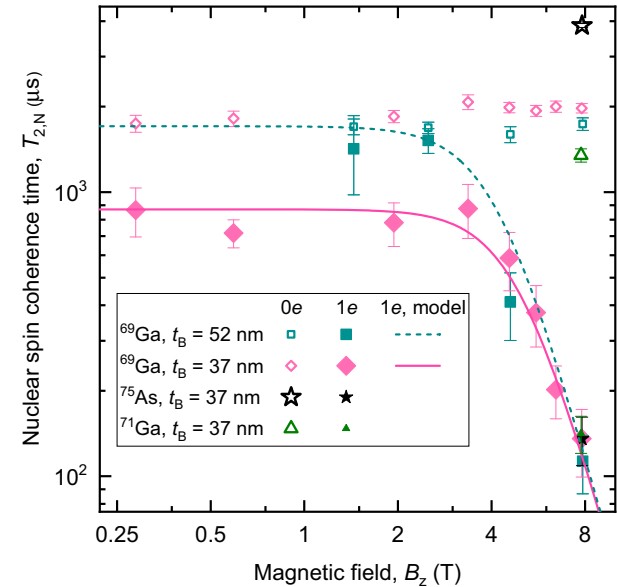

**Fig. 2 Coherence time of the nuclear spins coupled to quantum dot electron spin qubit. a** Gate bias ($V_G$) dependence of the $^{69}$Ga nuclear spin echo decay time $T_{2,N}$ in the $t_B = 52$ nm sample at $B_z = 1.94$ T (circles) and $B_z = 7.8$ T (squares). Solid lines are a guide to the eye. Resonant cotunneling between QD and the Fermi reservoir leads to reduction of $T_{2,N}$ down to dephasing time $\approx T_{2,N}^{*,(1e)}$ (dashed line), observed as dips at $V_G \approx 0.33$ V and $\approx 0.5$ V, which separate the Coulomb blockade plateaus corresponding to stable QD charge states 0e, 1e, and 2e. **b** Nuclear spin coherence in an empty QD ($T_{2,N}^{(0e)}$, open symbols) and electron-charged QD ($T_{2,N}^{(1e)}$ solid symbols) measured as a function of magnetic field $B_z$ for spin-3/2 isotopes in samples with different $t_B$. Lines show $T_{2,N}^{(1e)}$ of $^{69}$Ga calculated according to the model of Eq. (1) for $t_B = 37$ nm (solid line) and $t_B = 52$ nm (dashed line) samples. Error bars are 95% confidence intervals.

confirming that QD is in single-electron (1e) Coulomb blockade. Nuclear coherence time $T_{2,N}^{(1e)}$ in this regime is measured as a function of magnetic field $B_z$ (solid symbols in Fig. 2b). At low fields, $T_{2,N}^{(1e)}$ is nearly independent of $B_z$, but changes over to $T_{2,N}^{(1e)} \propto B_z^{-4}$ scaling at $B_z \gtrsim 4$ T, in stark contrast to $T_{2,N}^{(1e)} \propto B_z^2 A^{-3}$ law predicted previously for hyperfine-mediated nuclear spin decoherence mechanism[16]. Furthermore, at high field $B_z \approx 7.8$ T we find similar $T_{2,N}^{(1e)} \approx 130$ μs for $^{75}$As and $^{71}$Ga nuclei, despite a factor of $\approx 1.8$ difference in their hyperfine constants $A$.

**Decoherence driven by spectral diffusion**. To explain the observed $T_{2,N}^{(1e)}$ values we invoke a model, historically known as "spectral diffusion"[24,34] in solid-state magnetic resonance (not to be confused with physical diffusion of spins in gases and liquids). In this model, one type of spins is observed, while the second type is a source of fluctuating local fields. In our case, the nuclear spins are observed, while the single-electron spin is the fluctuating environment. The evolution of the electron spin is intertwined with the nuclear spin dynamics[35], but for the present case, an adequate description is achieved by assuming that the electron spin makes uncorrelated random jumps between the $s_z = \pm 1/2$ states as sketched in Fig. 1b. The validity of this telegraph-process approximation comes from the fact that the nuclear spin phase is a time-integral of the electron spin evolution $s_z(t)$ (see Supplementary Note 2). As a result, only the low-frequency components play a role, and all the relevant information about $s_z(t)$ is captured by the rate of the electron spin flips. In the studied range of magnetic fields $B_z \leq 8$ T this rate $1/(2T_{1,e}) \lesssim 6000$ s$^{-1}$ (ref. [33]) is small compared to root mean square Knight shift $\delta\nu_e \approx 30$ kHz, derived from NMR spectra of Fig. 1e. In this regime of slow fluctuations, the nuclear spin decoherence time due to spectral diffusion approximately equals the electron spin lifetime $T_{2,N,SD} \approx 1.38 T_{1,e}$ (see derivation in Supplementary Note 2). Spectral diffusion dominates nuclear spin decoherence at high magnetic fields where electron spin lifetimes shorten as $T_{1,e} \propto B_z^{-4}$. In an empty QD nuclear dipole-dipole interaction causes nuclear spin ensemble decoherence on a timescale $T_{2,N}^{(0e)}$—including this mechanism, we find the following approximation for nuclear spin coherence time in presence of the electron central spin:

$$1/T_{2,N}^{(1e)} = 1/T_{2,N}^{(0e)} + 1/(1.38 T_{1,e}). \tag{1}$$

This closed form model fully describes the experimental dependence of $T_{2,N}^{(1e)}$ on $B_z$ (lines in Fig. 2b), as well as isotope-independent $T_{2,N}^{(1e)}$ at high $B_z$. In the $t_B = 52$ nm sample, $T_{1,e}$ is very long at $B_z \lesssim 3$ T, making the dipole-dipole mechanism dominant – and indeed we find $T_{2,N}^{(1e)} \approx T_{2,N}^{(0e)}$ in that case.

Qualitative understanding of spectral diffusion is found by considering two scenarios: In the absence of electron spin flips, the static inhomogeneous Knight shifts are fully refocused to form nuclear spin echo. By contrast, even a single-electron spin-flip unbalances the phases acquired by the nuclei before and after the refocusing $\pi$ pulse (except for those rare flips occurring within a short interval $\approx T_{2,N}^{*,(1e)}$ at the start or the end or the spin echo sequence). The probability to have zero electron flips in an increasing time interval decreases exponentially, and with it decays exponentially the average nuclear spin coherence on a timescale $T_{2,N,SD} \approx 1.38 T_{1,e}$. A direct observation of this process is shown in Fig. 3a, where we plot histograms of the spin echo amplitude $\Delta E_{hf}$ detected with single-shot probe pulses, as opposed to averaging over multiple pump-rf-probe cycles used for the data of Figs. 1 and 2. At long evolution times $\tau_{evol} \approx 1300$ μs the echo is destroyed by decoherence, resulting in a single peak at $\Delta E_{hf} = 0$. At short $\tau_{evol} \approx 0.4$ μs the nuclear coherence is preserved in most cases, leading to a peak at $\Delta E_{hf} \approx 1.9$ μeV. At intermediate $\tau_{evol}$ a bimodal distribution is observed, demonstrating the two discrete possibilities of nuclear spin echo preservation or destruction, if electron spin does not or does flip, respectively.

**Single-shot readout of the central spin qubit**. Figures 3b, c show single-shot NMR measurement with a sequence $(\pi/2)_x - \tau_{evol} - (\pi/2)_y$, which generates collective nuclear spin coherence and probes its quadrature component $\Delta E_{hf}$ following the free evolution time $\tau_{evol}$. In an empty QD (0e, Fig. 3b) the distribution of detected $\Delta E_{hf}$ is

unimodal with a width given by the optical readout noise. By contrast, a clear bimodal distribution is observed with a single electron in a QD (1e, Fig. 3c) at $\tau_{evol} \approx 0.3$ μs. The two modes correspond to the two discrete electron spin qubit states $s_z = \pm 1/2$, that add positive or negative phase $\propto \tau_{evol}$ to the nuclear spin coherence. The free evolution can be seen as a quantum logic gate with unitary propagator $= |\uparrow\rangle\langle\uparrow| \otimes \bigotimes_{j=1}^{N} e^{+\frac{iA_j\tau_{evol}}{\hbar}\hat{I}_{z,j}} + |\downarrow\rangle\langle\downarrow| \otimes \bigotimes_{j=1}^{N} e^{-\frac{iA_j\tau_{evol}}{\hbar}\hat{I}_{z,j}}$, where the electron spin state $\uparrow$ or $\downarrow$ controls the phase of the nuclei ($\otimes$ is the Kronecker product operator). Owing to the inhomogeneity of the QD electron-nuclear spin coupling, the phase acquired by the $j$-th individual nuclear spin ($1 \leq j \leq N$) is proportional to its hyperfine constant $\propto A_j$. The final $(\pi/2)_y$ pulse transforms this phase into the optically detected hyperfine shift, thus implementing a single-shot readout of the electron spin qubit, which is a key ingredient in quantum information processing.

In order to characterize the noise in the single-shot readout, we use the double Gaussian model[36,37] for the distribution of the optically measured hyperfine shifts $\Delta E_{hf}$ (lines in Fig. 3c). Setting the threshold level in the middle between the two Gaussian modes ($\Delta E_{hf} = 0$ in Fig. 3c), we interpret the above-threshold readouts $\Delta E_{hf} > 0$ as electron spin up $s_z = +1/2$, and $\Delta E_{hf} < 0$ as electron spin down $s_z = -1/2$. Using the fitted parameters, we calculate $\gtrsim 99.8\%$ for the probability that the single-shot measurement of $\Delta E_{hf}$, including Gaussian noise, gives the correct electron spin state $s_z$ (see details in Methods). Previously, single-shot readout of electron spin state $s_z$ in an optically active QD was achieved by measuring fluorescence under resonant optical excitation, which acts back on the electron spin, limiting the readout fidelity. Our approach is different, in that it first converts the fragile electron spin degree of freedom into a more robust nuclear spin ensemble polarization, which we then readout optically. The conversion approach is known to provide improved readout fidelity, for example by transforming spin into charge, as demonstrated in gate-defined quantum dots[26,31], and more recently in diamond spins[32].

There are two factors limiting the overall readout speed and fidelity. One is the noise in the optical signal, which requires detection time on the order of 1–10 ms to achieve the estimated above single-shot accuracy of $\gtrsim 0.998$. These parameters can be enhanced further by optimizing the spectral resolution and collection efficiency of the optical detection setup. The other factor is related to the finite lifetime of the electron spin qubit. The flips of the electron spin during its conversion into the quadrature nuclear spin polarization impose a fidelity limit on the order of $\approx 1 - (T_{2,N}^{*,(1e)}/T_{1,e})$, where the microsecond-range nuclear spin dephasing time $T_{2,N}^{*,(1e)}$ is also the optimal conversion time. In the particular experiment of Fig. 3c, conducted at $B_z = 7.8$ T, we estimate the spin-flip fidelity limit to be $\approx 0.99$, although this bound can be increased to $>0.9999$ by simply reducing the field to $B_z \lesssim 2$ T, where the electron spin lifetimes exceed $T_{1,e} > 10$ ms (see Methods). Taking the worst of the two limiting factors, we arrive to an overall fidelity estimate $>0.99$, comparable to the state of the art in spin[26,27,30–32] and superconducting qubits[28,29]. Notably, the single-shot readout is achieved using a small fraction (<13%, see Methods) of the QD nuclei, so that the remaining nuclear spins can be used as a quantum resource. For example, it is possible to store multiple snapshots of the electron spin qubit state, by converting it into different nuclear isotope states ($^{69}$Ga, $^{71}$Ga, $^{75}$As), followed by the optical retrieval at the end of operations on the qubit.

The relative weights of the two modes in Fig. 3c reveal the electron spin polarization degree $\rho_e \approx 0.35$, corresponding to equilibrium of an electron with $g_e \approx -0.63$, in good agreement with electron g-factors found in similar QDs[38]. At long evolution times $\tau_{evol} \gg T_{2,N}^{*,(1e)}$ the coherence imprinted by the electron spin

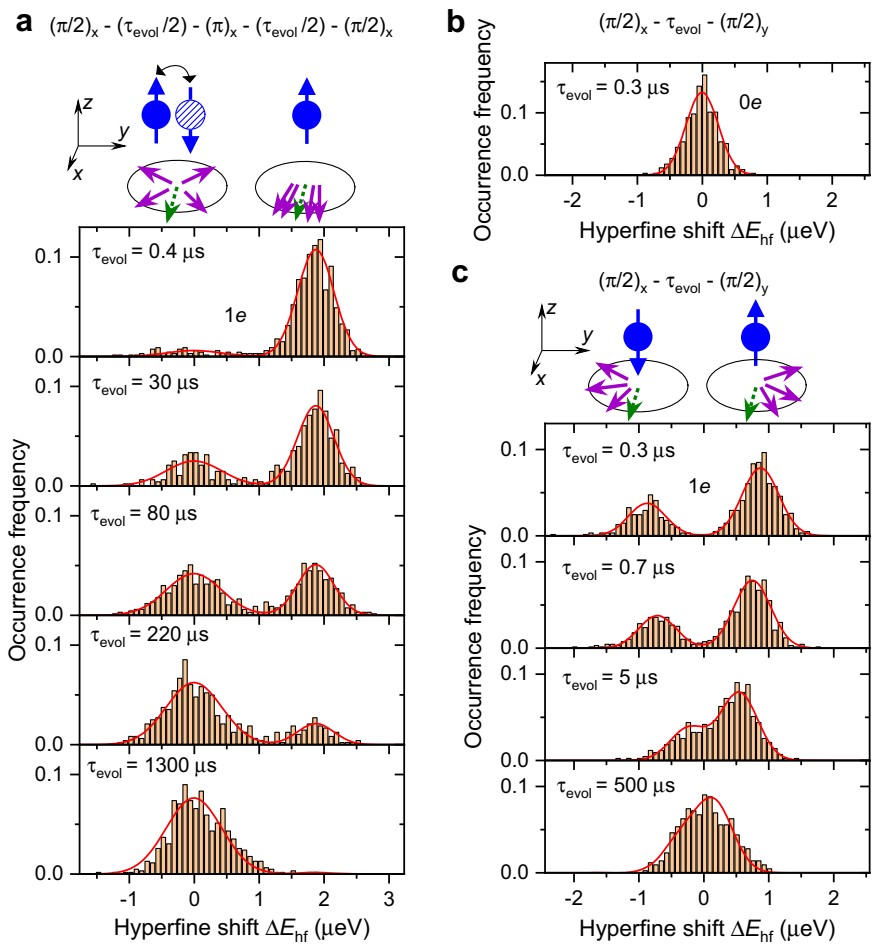

**Fig. 3 Single-shot NMR detection of the electron spin state. a** Spin echo of $^{75}$As nuclei in a $t_B = 37$ nm sample at $B_z \approx 7.8$ T measured with a single-shot optical detection of the resulting hyperfine shift variation $\Delta E_{hf}$. Measurement rf pulse sequence is $(\pi/2)_x - (\tau_{evol}/2) - (\pi)_x - (\tau_{evol}/2) - (\pi/2)_x$ with several values of the total free evolution time $\tau_{evol} = 0.4 - 1300$ μs shown. Results are plotted as histograms of the detected single-shot spin echo amplitudes $\Delta E_{hf}$. Lines show double Gaussian fits. Schematic shows electron spin in an $s_z = +1/2$ or $-1/2$ states (balls with up or down arrows). After the initial $(\pi/2)_x$ pulse all nuclear spins point along the same axis orthogonal to $z$ (dashed small arrow) and then precess around the $z$ axis to point along generally different directions (solid small arrows) prior to the final $(\pi/2)_x$ pulse: An electron spin-flip during the nuclear spin precession dephases the spins, resulting in $\Delta E_{hf} \approx 0$, whereas in the absence of electron flips, nuclear spin echo is formed, resulting in $\Delta E_{hf} \approx 1.9$ μeV. **b** Single-shot measurement of the free induction decay in an empty QD (0e), using sequence $(\pi/2)_x - (\tau_{evol}) - (\pi/2)_y$, where subscripts $x, y$ denote the equatorial axes in the rotating frame towards which the spins are flipped by the rf pulses. Line shows Gaussian fit. **c** Same sequence as in (**b**) applied to a charged QD (1e). For a sufficiently short evolution time $\tau_{evol} \lesssim T_{2,N}^{*,(1e)}$ the spin $s_z = -1/2$ ($s_z = +1/2$) of a single-electron pointing down (up) gives rise to a negative (positive) quadruature nuclear spin polarization $\Delta E_{hf}$, observed as a bimodal distribution of the single-shot-detected $\Delta E_{hf}$. Lines show double Gaussian fits.

is dephased, resulting in a unimodal distribution around $\Delta E_{hf} \approx 0$ (e.g., at $\tau_{evol} \approx 500$ μs).

**Dynamical decoupling of the spin ensemble.** Spin echo sequence with a single refocusing $\pi$ pulse is the simplest case of a more general concept of dynamical decoupling[39], where fast (bang-bang) rf control pulses are applied to filter out the unwanted interactions and increase the coherence time. For a single spin, decoupling from external environment can be achieved with a sequence of $\pi$ pulses[40,41], whereas for nuclear spin ensembles, sequences of $\pi/2$ and $\pi$ pulses, such as CHASE[22], have been designed to suppress both the ensemble inhomogeneity and the internal dipolar interactions. We examine $^{69}$Ga nuclear spin coherence under dynamical decoupling in two regimes of magnetic field.

At low magnetic fields ($B_z = 2.5$ T), where electron-induced spectral diffusion is weak, we find that a 5-pulse sequence CHASE-5 provides an improved coherence $T_{2,N}^{(1e)} \approx 2.2$ ms, compared to a single-pulse spin echo $T_{2,N}^{(1e)} \approx 1.5$ ms. This confirms the validity of

the dynamical decoupling approach for a spin ensemble inhomogeneously coupled to the central spin. Dynamical decoupling sequences are designed such that higher control rate (larger number of control pulses) converges the residual Hamiltonian to zero, progressively increasing the coherence time $T_{2,N}$. Experiments with longer decoupling sequences CHASE-10 and CHASE-20 show reduction of the echo amplitude to a level where $T_{2,N}$ becomes unmeasurable (see Supplementary Note 5), which we ascribe to strong inhomogeneous Knight shift broadening. However, this limitation is technical, and can be addressed by increasing the rf control field amplitude and bandwidth. Since nuclear spin decoherence in both neutral (0e) and charged (1e) QDs is governed by the same mechanisms of inhomogeneous broadening and nuclear dipolar interactions, we expect that $T_{2,N}^{(1e)}$ can be at least as large as $T_{2,N}^{(0e)} > 10$ ms, as found previously in neutral QDs[9,22]. Thus, dynamical decoupling can be a route to long coherence storage in many-body ensembles, with coherence times comparable to dilute individual nuclear spins[5].

Experiments at high magnetic field ($B_z = 7.8$ T), where nuclear spin coherence is dominated by the random electron spin flips, show that CHASE dynamical decoupling no longer provides any improvement over spin echo (see data in Supplementary Note 5). The decoherence mechanism is similar to the case of a single-pulse spin echo. A single-electron spin-flip is sufficient to completely destroy the balance of phases accumulated by spins in the free evolution intervals of the decoupling sequence. As a result, nuclear spin coherence is essentially limited by the electron spin lifetime $T_{2,N}^{(1e)} \lesssim 1.38 T_{1,e}$, and dynamical decoupling at high fields would require the control rate to be high enough to perform multiple decoupling cycles on a timescale shorter than $T_{1,e}$. A more feasible approach would be to operate in the low-field regime, where electron spin is essentially static. In this case, optimal coherence of the nuclear spin environment would be achieved by synchronizing its dynamical decoupling with the coherent control of the electron central spin qubit.

## Discussion

Spectral diffusion was previously observed for individual electron spins subject to fluctuations of the dilute nuclear spin ensembles[11]. Our results show that the same concept works in reverse, and describes the decoherence of 100% abundant nuclear spin ensembles subject to fluctuations of a single electron in a III–V semiconductor quantum dot. We show experimentally that many-body interactions and inhomogeneous coupling to the common central spin are not obstacles to long coherence storage in a nuclear spin ensemble, which is required to implement recent proposals for QD spin–photon networks with nuclear spin quantum memories[7] and registers[9]. In contrast to previous studies[16], we find no signature of decoherence arising from indirect hyperfine-mediated nuclear-nuclear interactions, which are expected to be prominent only below a certain magnetic field. Earlier works, not related to nuclear spin coherence, reported varying upper-boundary values $0.02 - 0.75$ T, for the range of magnetic fields where hyperfine-mediated interactions play an important role[42–44]. Our results show that nuclear spin coherence is unaffected by hyperfine-mediated nuclear-nuclear couplings at least down to $B_z \approx 0.25$ T, offering a wide range of magnetic fields suitable for long coherence storage.

While our experiments are conducted on collective multi-quantum coherent excitations of the nuclear spin ensemble, we expect the same mechanisms and similarly long coherence times for the single-quantum spin-wave excitations. Generation of single nuclear spin waves (single magnons) in quantum dots has been reported recently[6], but with short dephasing time (few microseconds). Our results suggest that once inhomogeneity is refocused, the coherence of the spin-wave quantum memories in quantum dots may be extended to milliseconds. The findings of this work on InGaAs/GaAs QDs are expected to apply to high-quality optically active GaAs/AlGaAs QDs, where small intrinsic strain[9] holds a promise for even longer coherence and complete control of the hybrid electron-nuclear spin quantum system.

We have demonstrated generation of the collective nuclear spin phase, controlled by the electron spin qubit state. The fidelity of this operation is limited by inhomogeneity of the electron-nuclear interactions. This problem can be addressed by operating on small subensembles of nuclear spins with similar hyperfine coupling to the electron spin. Such manipulation can be achieved with spectrally selective control pulses[45]—this approach is feasible with the high-sensitivity optical techniques, capable of detecting a few-percent fraction of the nuclear spin ensemble, as demonstrated in our experiments. Even without any additional development, we demonstrate that controlled-phase operation on a small fraction (<13%) of nuclear spins can be used for a single-

shot high-fidelity readout of the electron spin qubit. The remaining nuclei can be used as a spin-wave quantum memory. Alternatively, the QD nuclear spins can be divided into small subensembles, e.g., by the isotope type ($^{69}$Ga, $^{71}$Ga or $^{75}$As), and used to record the electron spin state at multiple timepoints. This for example, can alleviate the need to initialize the electron spin qubit prior to its coherent manipulation—both the initial and final electron spin states can be stored in the nuclei for subsequent optical retrieval. Future work will include exploration of the new opportunities offered by a highly coherent electron-nuclear spin system in individual quantum dots.

## Methods

**Sample structures.** The samples are low-density InAs self-assembled QDs grown on a GaAs substrate using molecular beam epitaxy. The dots are placed between two distributed Bragg reflectors consisting of GaAs and AlAs layers and forming a $\lambda/2$ optical cavity. The low temperature ground-state optical emission of the studied QDs is at $\approx 950$ nm. The Fermi reservoir is a Si doped GaAs layer (Si concentration $\approx 1.1 \times 10^{18}$ cm$^{-3}$), separated from QDs by a GaAs tunnel barrier layer of thickness $t_B$. The samples are processed into planar Schottky diode structure, allowing for the charge state of the dots to be controlled by applying external bias $V_G$ to the top metal gate. The studied semiconductor device have been examined previously using photoluminescence, resonance fluorescence and spin lifetime measurement techniques—the detailed results can be found in ref. [33] and the Supplementary Information therein.

Electron spin relaxation rates $\Gamma_e = 1/T_{1,e}$ measured at low temperature $T = 4.5$ K are well described by the following model[33]:

$$\Gamma_e = \Gamma_{e,cotun} + \Gamma_{e,ph} B_z^{k_{ph}} \qquad (2)$$

where the first term $\Gamma_{e,cotun}$ accounts for the field-independent relaxation induced by electron cotunneling and the second term describes the field-dependent relaxation induced by acoustic phonons. Equation (2) is then substituted into Eq. (1) to find a closed form dependence of the nuclear spin coherence time on the external magnetic field $B_z$. The phonon mechanism parameters are $\Gamma_{e,ph} \approx 2.27 \pm 0.48$ s$^{-1} \times T^{-k_{ph}}$ and $k_{ph} \approx 4.1 \pm 0.13$ in both samples[33]. Cotunneling depends on the barrier thickness. For the $t_B = 52$ nm sample the range of values is $1/\Gamma_{e,cotun} \approx 1.26 - 1.65$ s for different individual QDs. For the $t_B = 37$nm sample the dot-to-dot variation is more pronounced $1/\Gamma_{e,cotun} \approx 1.2 - 5.8$ ms. Electron spin lifetimes have not been measured for the particular QDs used here for nuclear spin coherence studies, and thus we treat $\Gamma_{e,cotun}$ in the $t_B = 37$ nm sample as the only adjustable parameter. The best fit value plotted by the solid line in Fig. 2b is $1/\Gamma_{e,cotun} \approx 0.96$ ms, in good agreement with the range of rates found from direct measurements on other QDs in the same sample.

**Experimental techniques.** The sample is placed in a liquid helium bath cryostat equipped with a superconducting magnet, providing a field up to $B_z = 8$ T, parallel to sample growth direction and optical axis $z$ (Faraday geometry). An aspheric lens is used for optical excitation of the QD and photoluminescence (PL) collection. Diode lasers emitting at 850 nm are used both for optical polarization of the nuclear spins (optical pump pulses) and PL excitation (optical probe pulses). The collected PL is dispersed with a 1 m double grating spectrometer and recorded with a charge-coupled device camera. The changes in the spectral splitting of a negatively charged trion $X^-$, derived from the PL spectra, are used to measure the hyperfine shifts $E_{hf}$ proportional to the nuclear spin polarization degree (see Supplementary Fig. 3). The oscillating magnetic field $B_x \perp z$ implementing the coherent control of the collective nuclear spin state is produced by a coil placed at a distance of $\approx 0.5$ mm from the QD sample. The coil is made of 10 turns of a 0.1 mm diameter enameled copper wire wound on a $\approx 0.4$ mm diameter spool in five layers, with two turns in each layer. The coil is fed by a 1 kW rf amplifier through a resonant impedance matching network made of 50 $\Omega$ coaxial cables.

**Optically detected NMR.** The details of NMR implementation are given in Supplementary Note 3. Overall the experimental cycle follows the timing diagram shown in Fig. 1b. As in previous work[16,22,46], the optically induced nuclear spin state is augmented with adiabatic rf frequency sweeps over the inhomogeneously broadened satellite NMR transitions $-3/2 \leftrightarrow -1/2$ and $+1/2 \leftrightarrow +3/2$. The sweeps exchange the populations of the $I_z = -3/2$ and $-1/2$ pair of states, as well as populations of the $I_z = +1/2$ and $+3/2$ pair. This way the population difference of the $I_z = \pm 1/2$ pair is maximized, prior to coherent manipulation of the $I_z = \pm 1/2$ subspace. The amplitude of the frequency-swept rf excitation is chosen to produce Rabi frequency between $\approx 1 - 4$ kHz, and the typical sweep rates are between $5-10$ MHz s$^{-1}$. In some experiments, such as single-shot measurements of Fig. 3, the same adiabatic sweeps are applied for the second time, after the coherent manipulation, transferring the final populations of the $I_z = \pm 1/2$ states back into $I_z = \pm 3/2$ states. This gives a factor of $\approx 3$ increase in the variation of the optically detected hyperfine shifts $\Delta E_{hf}$ at the expense of a longer experimental time. In the case such second set of sweeps is used, all the measured $\Delta E_{hf}$ are divided by 3 to obtain the values directly comparable

with the measurements where adiabatic sweeps are applied only after the optical pumping. In all experiments, except for single-shot measurements, the hyperfine shift $\Delta E_{hf}$ is acquired by averaging over $15-60$ pump-rf-probe cycles shown in Fig. 1b to obtain an approximation to a statistical average NMR signal. In echo decay experiments, such as shown in Fig. 1f, the dependence of the echo amplitude on the total free evolution time $\tau_{evol}$ is fitted with stretched or compressed exponentials $\propto e^{-(\tau_{evol}/T_{2,N})^{\eta}}$, where $\eta$ is the parameter describing stretching ($\eta < 1$) or compression ($\eta > 1$).

Coherent control of the nuclear spins is achieved using high power rf pulse bursts with rotating-frame amplitude of up to $B_1 \approx 10$ mT, which corresponds to laboratory-frame amplitude $B_x \approx 20$ mT and $^{75}$As Rabi frequency $\nu_1 = 2\gamma B_1/(2\pi) \approx 140$ kHz, where the additional factor of 2 is from the matrix element of the spin-3/2 operator projected onto the $I_z = \pm 1/2$ subspace. Owing to the internal strain of the QDs, the quadrupolar shifts of the $I_z = \pm 3/2$ nuclear spin states are in MHz range, much larger than $\nu_1$. As a result, the $I_z = \pm 3/2$ states are strongly detuned, making it possible to perform selective coherent rotations within the $I_z = \pm 1/2$ subspace of interest. In order to achieve broadband uniform rotation of the spin ensemble, $\nu_1$ must be larger than the resonance spectral broadening of the $I_z = \pm 1/2$ subspace. While larger $B_1$ can be achieved by increasing the rf power, the practical limitations arise from the parasitic effects of the rf electric field. Above certain level, typically corresponding to $B_1 \approx 5$ mT, high power rf pulses are found to induce electron spin flips, which then disrupt the formation of the nuclear spin echo. In single-shot spin echo experiments with short $\tau_{evol}$ the rf-induced electron spin flips are observed as nearly equal weights of the two modes in the histogram. Consequently, experiments shown in Fig. 3a are conducted at a reduced $B_1 \approx 5$ mT, where the probability of parasitic electron spin flipping is estimated from fitting to be within <0.05. The downside of a low $B_1$ is the reduced spectral bandwidth of the control pulses, which leads to incomplete rotation for some of the spins and a reduced spin echo amplitude. In those spin echo experiments where the readout is averaged over multiple pump-probe cycles, the contributions of the cycles where electron spin is flipped by the rf field can be ignored, leading to correct spin echo decay time $T_{2,N}^{(1e)}$ but with a reduced echo amplitude. Future work may include optimization of rf circuitry with the aim of maximizing $B_1$ while reducing the parasitic rf electric field.

Most of the pulsed nuclear magnetic resonance experiments are conducted on the $^{69}$Ga isotope due to its favorable balance between the hyperfine shift amplitude, Knight shift inhomogeneity, and the quadrupolar inhomogeneity. Additional results for $^{75}$As isotope can be found in Supplementary Note 4.

**Estimate of the number of nuclei used for single-shot electron spin detection**. The single-shot NMR experiments presented in Fig. 3 are conducted on $^{75}$As nuclei, which have 100% natural abundance and thus make up 50% of all the nuclei in a QD, the rest being the group-III Ga, In and possibly some Al nuclei. The nuclear spin polarization degree produced by optical pumping is estimated to be $|\rho_N| \approx 0.65$ (refs. 47,48), which corresponds to a dimensionless inverse spin temperature $|\beta| \approx 0.98$ in a Boltzmann distribution of the nuclear spin level population probabilities $p(I_z) \propto \exp(\beta I_z)$. The nuclei that are initially in the $I_z = \pm 3/2$ states are transferred into the $I_z = \pm 1/2$ states by adiabatic rf frequency sweeps—only these nuclei contribute to NMR signal, and from the $p(I_z)$ distribution their fraction is ≈0.67. Finally, the reduced rf amplitude used in single-shot NMR ($^{75}$As CT Rabi frequency $\nu_1 \approx 70$ kHz) results in selective rf pulses, which excite only the nuclei whose NMR frequencies are within the pulse bandwidth $\propto \nu_1$. In spin echo experiments, such lowering of the rf amplitude results in spin echo amplitude that is a factor of ≈0.38 smaller compared to the echo amplitude in experiments conducted with the highest possible rf amplitude. Combining all these factors we find an upper estimate $<0.5 \times 0.67 \times 0.38 \approx 0.13$ of the fraction of the QD nuclei taking part in a single-shot detection of the electron spin state. With the total number of nuclei estimated to be $N \approx 4 \times 10^4$ in the studied dots[33], this corresponds to ≈5000 nuclei participating in electron spin detection.

**Estimate of the single-shot electron spin readout fidelity and its limiting factors**. The histograms of the single-shot quadrature NMR signal $\Delta E_{hf}$ shown in Fig. 3c are fitted with a double Gaussian function $A_{(-)}2^{-\left(\frac{\Delta E_{hf}+\Delta E_{hf,0}/2}{w/2}\right)^2} + A_{(+)}2^{-\left(\frac{\Delta E_{hf}-\Delta E_{hf,0}/2}{w/2}\right)^2}$, where $\Delta E_{hf,0}$ is the splitting of the two modes, $w$ is the full width at half maximum of each mode peak and $A_{(+)}$ ($A_{(-)}$) is the amplitude of the mode corresponding to positive (negative) average hyperfine shift variation $\Delta E_{hf}$. The relative difference of the amplitudes reflects the electron spin polarization degree $\rho_e = (A_{(+)} - A_{(-)})/(A_{(+)} + A_{(-)})$. For the measurement at $\tau_{evol} = 0.3$ μs in Fig. 3c we find $\Delta E_{hf,0} \approx$ 1.75 μeV, and the width $w \approx 0.67$ μeV determined by collection efficiency and spectral resolution of the instruments used to analyze QD PL. When the quadrature NMR signal $\Delta E_{hf}$ is measured, it is interpreted as electron spin state $s_z = +1/2$ ($s_z = -1/2$ for $\Delta E_{hf} > 0$ ($\Delta E_{hf} < 0$). Thus, the total probability of correct detection is

$$F = \left( (A_{(-)} + A_{(+)}) \int_{-\infty}^{\infty} 2^{-\left(\frac{x}{w/2}\right)^2} dx \right)^{-1}$$
$$\times \left( \int_{-\infty}^{0} A_{(-)} 2^{-\left(\frac{x+\Delta E_{hf,0}/2}{w/2}\right)^2} dx \right.$$
$$\left. + \int_{0}^{\infty} A_{(+)} 2^{-\left(\frac{x-\Delta E_{hf,0}/2}{w/2}\right)^2} dx \right).$$

Taking the integrals we find for the readout fidelity:

$$F = \frac{1}{2}\left( 1 + \text{erf}\left[ \frac{\sqrt{\ln 2}\,\Delta E_{hf,0}}{w} \right] \right), \qquad (3)$$

where erf is the standard Gauss error function. Evaluating this for the measurement with $\tau_{evol} = 0.3$ μs, we find $F \approx 0.9989$. This is the value quoted in the main text above and it is determined by the accuracy with which the collective nuclear spin polarization of the quantum dot is readout optically. Increasing the optical probe duration $T_{Probe}$ increases the number of the PL photons collected and reduces $w$, thus improving the measurement accuracy of the spectral splitting variation $\Delta E_{hf}$. However, if $T_{Probe}$ is too long, it depolarizes the nuclei and reduces the separation $\Delta E_{hf,0}$ of the histogram modes, thus reducing the readout fidelity. Therefore, there is an optimal probe duration $T_{Probe}$, which maximizes the fidelity $F$. In our experiments $T_{Probe}$ is set to this optimal value. Further improvement of the electron spin readout fidelity can be achieved using, e.g., solid immersion lenses[49] in order to increase probe PL photon collection without the need for a longer $T_{Probe}$.

Even for a perfect (noise-free) optical probing, the readout fidelity is limited by the flips of the electron spin during its conversion into the nuclear spin coherence. The conversion takes place both during the free evolution time $\tau_{evol}$, and partially during the $\pi/2$ control pulses of the $(\pi/2)_x - \tau_{evol} - (\pi/2)_y$ sequence. Simple calculations show that for any realistic distribution of the inhomogeneous Knight shifts, the peak amplitude of the quadrature signal generated by the $(\pi/2)_x - \tau_{evol} - (\pi/2)_y$ sequence is reached when the evolution time approximately equals the dephasing time $\tau_{evol} \approx T_{2,N}^{*,(1e)}$. However, in the experiment (Fig. 3c), the quadrature signal (splitting $\Delta E_{hf,0}$ of the modes) is maximized at $\tau_{evol} = 0.3$ μs, much shorter than $T_{2,N}^{*,(1e)} \approx 4.5$ μs. This suggests that some conversion takes place during the control pulses, whose duration is $t_{\pi/2} = 3.75$ μs. Owing to the spin locking[50] arising when the nuclear spin control pulses are strong (i.e., Rabi frequency of the rf pulse $\nu_1 \approx 70$ kHz is sufficiently large compared to Knight shift broadening $\delta\nu_e \approx 30$ kHz), the effect of the electron spin flips during rf excitation is smaller than during the free evolution. Thus, we estimate that the effective conversion time $\tau_{conv}$ in our experiment is longer than $\tau_{evol} = 0.3$ μs, but does not exceed $T_{2,N}^{*,(1e)} \approx 4.5$ μs. By increasing the rf field amplitude $B_1$, one can achieve $\tau_{conv} \approx \tau_{evol}$. The probability that the electron spin is preserved during its conversion into nuclear spin is $\approx 1 - (\tau_{conv}/T_{1e})$. Taking $\tau_{conv} \approx 1$ μs and $T_{1,e} \approx 90$ μs, measured previously[33] at high magnetic field $B_z = 7.8$ T, we find the fidelity upper bound of ≈0.99. Electron spin flips occurring at random times during the conversion result in quadrature nuclear spin polarization values spread over the entire range $[-\Delta E_{hf,0}/2, +\Delta E_{hf,0}/2]$. Such additional signal must appear as broad background in the histograms of Fig. 3c, but in practice is too small to be observed directly, explaining why the fidelity derived from Gaussian fitting exceeds the upper bound associated with electron spin flips. On the other hand, the role of the electron spin flips during the conversion stage of the readout can be suppressed if magnetic field is reduced to ≤2 T, where $T_{1,e} \geq 10$ ms (ref. 33), and the fidelity would be governed by the statistical noise in the optically detected hyperfine shift $\Delta E_{hf}$ as derived in the previous paragraph.

In summary, the overall fidelity of the electron spin readout is limited by the strongest of the two competing effects. One mechanism is due to the electron spin flips taking place during its conversion into the nuclear spin coherence—this factor is largely related to the quality of the electron spin qubit, and would affect any readout method. The resulting fundamental limit on fidelity $\approx 1 - (T_{2,N}^{*,(1e)}/T_{1,e})$ can exceed 0.9999 for QDs at reasonably low few-Tesla magnetic fields. The other mechanism is related to the second stage of the readout, where the nuclear spin coherence is retrieved optically. Here, optimized collection of the light emitted by the QD, can improve fidelity beyond the $\gtrsim 0.998$ level demonstrated in this work.

## Data availability

The data that support the findings of this study are available from the corresponding authors upon reasonable request.

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

## Acknowledgements

Royal Society provided funding support to E.A.C. through University Research Fellowship and grant RGF\EA\180117, and to G.G. and E.A.C through grant RG150465. The work was supported by EPSRC grant EP/V048333/1.

## Author contributions

G.G. and E.A.C developed experimental techniques, analyzed the data, and wrote the manuscript. G.G. conducted experiments. E.C. grew the samples. E.A.C. coordinated the project.

## Competing interests

The authors declare no competing interests.
