## [Peer Review File · Nature Communications]

REVIEWERS' COMMENTS

Reviewer #1 (Remarks to the Author):

In comparison to the manuscript submitted previously to NatPhys a large number of suggestions of the three referees has been considered and implemented. Thus, the manuscript is clearer and better focused on the obtained achievements. The interesting and relevant many-body system built from an electron spin and a large number of nuclear spins is significantly better understood due to the submitted study. Thus I recommend its publication as Nature Communication.

One thing should still be elucidated:

In the response to Reviewer 3 on page 9 the value of $T_{2,N}^*$ is estimated. First, it is simply computed from the maximum value of A_i , then from its root-mean-square. What do you think is the really relevant quantity?

Would a constant A_i avoid any dephasing?

Reviewer #3 (Remarks to the Author):

The authors have done a significant revision of their original manuscript.

I think they have addressed all the comments of the referees in a very convincing manner.

In my opinion, the article improved a lot in terms of readability and addressing the relevant research target.

I support the publication of this manuscript in its present form.

RESPONSE TO REVIEWERS' COMMENTS (2nd REVIEW)

Reviewer #1 (Remarks to the Author):

In comparison to the manuscript submitted previously to NatPhys a large number of suggestions of the three referees has been considered and implemented. Thus, the manuscript is clearer and better focused on the obtained achievements. The interesting and relevant many-body system built from an electron spin and a large number of nuclear spins is significantly better understood due to the submitted study. Thus I recommend its publication as Nature Communication.

One thing should still be elucidated: In the response to Reviewer 3 on page 9 the value of $T_{2,N}^$ is estimated. First, it is simply computed from the maximum value of A_i , then from its root-mean-square. What do you think is the really relevant quantity? Would a constant A_i avoid any dephasing?*

Indeed, a constant A_i for all nuclei in a quantum dot would dramatically reduce nuclear spin ensemble dephasing. In a real QD, the distribution of A_i is controlled by the shape of the electron envelope wavefunction. A constant A_i distribution would require the wavefunction to be rectangular (“box function”), which is not feasible.

Using $\max(A_i)$ gives a rough but straightforward estimate of the nuclear spin ensemble dephasing time $T_{2,N}^*$. The root-mean-square of A_i would give a more realistic estimate of the dephasing time $T_{2,N}^* \sim 2\hbar/\text{RMS}(A_i)$, whereas the most accurate calculation of the dephasing, which is not necessarily exponential and therefore cannot be described by a single $T_{2,N}^*$ parameter, must use the exact shape of the electron envelope wavefunction. Leaving aside the infeasible cases such as constant A_i , for any realistic electron wavefunction the ratio of max and RMS is a coefficient of order unity. For example, for a one dimensional Gaussian wavefunction the RMS of A_i is evaluated to be $\sim 0.439 \cdot \max(A_i)$, while for a uniform distribution of A_i the RMS is $\sim 0.577 \cdot \max(A_i)$. Even more importantly, for the practically relevant slow-electron-spin-flip regime, the exact distribution of A_i is not that important: for any realistic distribution of A_i the nuclear spin coherence time is $T_{2,N} \sim 1.38 \cdot T_{1,e}$, where $T_{1,e}$ is the electron spin lifetime. That is why we limit ourselves to a rough estimate of $T_{2,N}^*$, which is quoted in the introduction of the paper as $2\hbar/\text{RMS}(A_i)$.

While constant A_i is not possible for the entire QD, as we mention in the Discussion section, it might be possible to approach this limit by operating on small subensembles of nuclear spins with similar hyperfine coupling to the electron spin. Such manipulation can be achieved with spectrally-selective control pulses and may provide a route for extended nuclear spin dephasing times.

Reviewer #3 (Remarks to the Author):

The authors have done a significant revision of their original manuscript. I think they have addressed all the comments of the referees in a very convincing manner. In my opinion, the article improved a lot in terms of readability and addressing the relevant research target. I support the publication of this manuscript in its present form.

We would like to thank both Reviewers for their efforts in reviewing this manuscript and their positive recommendations.